# Identifying Risk Genes and Interpreting Pathogenesis for Parkinson’s Disease by a Multiomics Analysis

**DOI:** 10.3390/genes11091100

**Published:** 2020-09-21

**Authors:** Wen-Wen Cheng, Qiang Zhu, Hong-Yu Zhang

**Affiliations:** Hubei Key Laboratory of Agricultural Bioinformatics, College of Informatics, Huazhong Agricultural University, Wuhan 430070, China; chengww@webmail.hzau.edu.cn (W.-W.C.); zhy630@mail.hzau.edu.cn (H.-Y.Z.)

**Keywords:** Parkinson’s disease, multiomics, risk genes, pathogenesis

## Abstract

Genome-wide association studies (GWAS) have identified tens of genetic variants associated with Parkinson’s disease (PD). Nevertheless, the genes or DNA elements that affect traits through these genetic variations are usually undiscovered. This study was the first to combine meta-analysis GWAS data and expression data to identify PD risk genes. Four known genes, *CRHR1*, *KANSL1*, *NSF* and *LRRC37A*, and two new risk genes, *STX4* and *BST1*, were identified. Among them, *CRHR1* is a known drug target, indicating that hydrocortisone may become a potential drug for the treatment of PD. Furthermore, the potential pathogenesis of *CRHR1* and *LRRC37A* was explored by applying DNA methylation (DNAm) data, indicating a pathogenesis whereby the effect of a genetic variant on PD is mediated by genetic regulation of transcription through DNAm. Overall, this research identified the risk genes and pathogenesis that affect PD through genetic variants, which has significance for the diagnosis and treatment of PD.

## 1. Introduction

Parkinson’s disease (PD) is called the “undead cancer” and is the second most common progressive neurodegenerative disease affecting the elderly population [1]. The global prevalence of PD is expected to double from 6.2 million cases in 2015 to 12.9 million in 2040 [2]. The complexity of PD leads to severe challenges, such as the ambiguity of early diagnosis and later symptoms that are difficult to control [3]. Therefore, finding effective genetic markers and analyzing their pathogenesis have become the focus of future clinical research on PD.

Genome-wide association studies (GWAS) identified tens of genetic variants associated with PD, most of which are from patients of European ancestry, and there is relatively little knowledge about PD in other populations [3]. We have a limited understanding of the biological functions of the risk alleles that have been identified, although PD risk variants seem to be in close proximity to known PD genes and lysosomal-related genes. Recently, Nalls et al. examined the risk relationships between PD and other phenotypes based on these genetic variants [4]. However, the pathogenic mechanisms of these genetic variations remain obscure, especially how the variants influence the gene expression through methylation [5]. In fact, the determination of genotype–phenotype causality based on GWAS still faces many difficulties: (i) most of the mutation sites are located in noncoding regions of the genome and are not protein-coding sequences [5]; (ii) the linkage disequilibrium (LD) effect of adjacent single nucleotide polymorphisms (SNPs) will produce a noncausality between genotype and phenotype [6]; and (iii) the limitation of sample size reduces the statistical effect of GWAS. Rare and low frequency variations may not reach the statistical threshold [7,8].

In the past decade, researchers have made many efforts to study the genetic structure of human traits or diseases. The development of emerging technologies such as single-cell sequencing, high-throughput screening, and machine learning algorithms will enhance our understanding of genetic risks. Recently, Zhu et al. developed the summary data-based Mendelian randomization (SMR) and heterogeneity in dependent instruments (HEIDI) methods [9], which provide a way forward for identifying functional genes and potential functional elements of genetic variation. The fundamental idea is that GWAS identifies SNPs related to phenotypes, and expression quantitative trait locus (eQTL) analysis obtains SNPs related to gene expression and then obtains risk genes that affect phenotypes by SNPs [10]. SMR allows for the use of GWAS, eQTL and methylation quantitative trait locus (mQTL) from independent studies to aggregate data, so it can expand a sufficient sample size to enhance statistical capabilities [11].

In this study, by integrating GWAS, eQTL and mQTL, the risk genes of PD were identified, and their potential pathogenesis was analyzed. First, we integrated two sets of GWAS data based on meta-analysis and found that the data can reflect the pathological features of PD by using Gene Ontology (GO) enrichment. Second, SMR and HEIDI were used to combine the meta-analysis GWAS result with eQTL data to identify risk genes that affect PD through genetic variation. Last, this study combined GWAS, eQTL and mQTL data to analyze the pathogenesis of risk genes. To determine the functional elements that regulate gene expression, this paper used epigenetic annotation data from the Roadmap Epigenomics Mapping Consortium (REMC) to locate the function of DNAm sites [12,13]. In summary, this research identified the risk genes and potential pathogenesis that affect PD, which has significance for the diagnosis and treatment of PD.

## 2. Materials and Methods

### 2.1. GWAS Analysis of Whole-Genome Data from PPMI

The whole-genome sequencing (WGS) data used in this study were derived from 606 individuals (412 PD patients and 194 controls) in the Parkinson’s Progression Marker Initiative (PPMI) database [14]. We used PLINK v1.07 toolset [15] for a standard case–control association analysis with age and gender as covariates. In the quality control phase, we selected SNPs with detection rates greater than 95%, sample detection rates greater than 95%, minimum allele frequencies greater than 5 × 10^−2^, and Hardy–Weinberg equilibrium less than 1 × 10^−4^.

### 2.2. Meta-Analysis Based on Effect Size

Meta-analysis was used to integrate two sets of GWAS data, including the above-mentioned summary-GWAS results from PPMI and GWAS summary data from another group of 4238 PD patients and 4239 controls, respectively, and meta-analysis was used in METAL tool [16] using an effect-based approach. A total of more than 6 million SNPs were obtained by meta-analysis GWAS, of which 1678 significant SNPs (Bonferroni-corrected *p*-value < 7.83 × 10^−9^) were associated with PD.

### 2.3. Summary Data-Based Mendelian Randomization Analysis

For the identification of PD risk genes based on GWAS and eQTL using SMR analysis, y was the phenotype, x was the gene expression, z was the genetic variation, and $b_{xy}$ ($b_{xy}$ = $b_{zy}/b_{zx}$) represent a non-confounding effect of gene expression on PD. The effect of SNPs on gene expression ($b_{zx}$, with a positive or negative sign) was the effect value of eQTL, and the effect of SNPs on PD ($b_{zy}$) was the effect value of meta-analysis GWAS. FDR adjustment at *p*-value < 5 × 10^−2^ was the threshold of SMR. Simultaneously, this study also infers the positive or negative effect of genes on PD (*b_xy_* < 0 means negative effect, otherwise the opposite).

The HEIDI test can exclude causality caused by LD in SMR analysis and obtain pleiotropic mutation sites (i.e., causality between two genomes is caused by the same genetic mutation) [17]. This study used the Bonferroni correction *p*-value as the threshold for HEIDI. Therefore, for each significant eQTL probe detected by SMR, we detected the $b_{xy}$ of multiple SNPs in the eQTL region to exclude false-positive genes.

In addition, the QTL data used are the aggregate data compiled by Zhu et al. [9,10]. The original eQTL summary data from the Westra et al. study included an eQTL meta-analysis of 5311 samples from peripheral blood [18] and a summary of the meta-analysis of Europeans from the Brisbane System Genetics Study (*n* = 614) and the Losian Birth Cohorts of 1921 and 193,631 (*n* = 1366) [19,20]. After exclusion of probes in the MHC areas and threshold screening (*p*-value < 5 × 10^−8^), a total of 5967 eQTL probes and 73,973 DNAm probes were obtained.

### 2.4. Enrichment Test of Functional Categories

In the public data of the REMC database at http://compbio.mit.edu/roadmap/, we downloaded the epigenetic annotation files of 25 chromatin states of 23 blood cells in 127 epigenomics studies [12,13]. The 14 main functional categories were obtained by merging 25 chromatin states with functionally related annotation categories into a single functional category. These annotation files were used to locate the 14 main functional regions of each methylation probe. After functional location of the DNAm probe, we quantified the ratio of transcript-associated DNAm significant probes to all probes in 14 major functional categories. Finally, the chi-square test was used to test the significance and positive or negative effects of enrichment with these significant methylation probes in each domain.

## 3. Results

### 3.1. Screening SNPs of PD Based on Meta-Analysis GWAS Data

We performed a GWAS using WGS data from the PPMI database including 412 PD patients and 194 controls [14]. Age and gender were used as covariates to obtain GWAS summary data through PLINK [15]. There were 48 SNPs significantly related to PD (*p*-value range was from 1.25 × 10^−7^ to 9.21 × 10^−7^).

The effect-based meta-analysis was then used to integrate GWAS data from PPMI with other aggregated GWAS data, which included 4238 PD patients and 4239 controls [21]. The results of the meta-analysis for GWAS obtained more than 6 million SNPs, where 1678 SNPs (Bonferroni-corrected *p*-value < 0.05/6,385,696 = 7.83 × 10^−9^) were significantly associated with PD (Figure 1).

Mapping the significant SNPs to the gene domain of the hg38 reference genome in a 10 kb interval [22], the GO enrichment results showed that six biological functional regions were significantly enriched (Bonferroni-corrected *p*-value < 8 × 10^−3^, Appendix A). Four significantly enriched biologically functional regions have been reported to be associated with PD, including the axon [23], neuron-to-neuron synapse [24], asymmetric synapse [25] and gamma aminobutyric acid (GABA)-ergic synapse [26] (Figure 2). Therefore, these results indicate that the meta-analysis of GWAS data reflects the pathological features of PD.

### 3.2. Identifying PD Risk Genes Based on GWAS and eQTL Data

SMR was used to integrate more than 6 million SNPs of meta-analysis GWAS data and 5966 cis-eQTL probes. After false discovery rate (FDR) adjustment at *p*-value < 5 × 10^−2^, we found eight genes (tagged by 11 probes) in the genome-wide region (Table 1). For each gene that passed the SMR, the HEIDI test was used to exclude the genes with an LD effect, and six (Bonferroni-corrected P_HEIDI_ ≥ 0.006) pleiotropic causal genes were found. As shown in Table 1 and Figure 3, the six risk genes were *CRHR1*, *KANSL1*, *NSF*, *LRRC37A*, *STX4* and *BST1*. Importantly, there is evidence that *CRHR1* [27,28], *KANSL1* [27], *LRRC37A* [29] and *BST1* [30,31,32] are associated with PD. Notably, a recent genetic analysis involving 8725 PD patients and 17,079 controls found that *BST1* is a susceptibility gene for PD [33], and previous experiments in *BST1* knockout mice suggested that the gene might be beneficial in promoting treatment for PD [34]; the top negatively related gene *CRHR1*, as a target for a hydrocortisone drug for Cushing’s disease [35], is also known as the corticotropin releasing factor (CRF) receptor 1, and there is evidence that CRF and its receptors protect neurons [36,37]. In addition, animal experimental evidence shows that hydrocortisone can stimulate the expression of Parkin through the CREB pathway, and the induced expression of Parkin is the cause of its neuroprotective effect [38].

We also found two new genes (*STX4* and *BST1*) in the above analysis, which are not the closest annotated genes to GWAS-significant SNPs. The reliability of these two new risk genes was verified from two aspects. On the one hand, we performed a protein–protein interaction (PPI) network analysis using STRING v11.0 database [39] to explore the association of six risk genes with known PD genes. The known functional genes were from the PD-Gene database (http://www.pdgene.org/). The results showed that five of the six genes interacted with known genes, including the new *STX4* gene (Figure 4b). On the other hand, the independent GWAS summary data from the latest analyses of PD were analyzed, and we found that *STX4* and *BST1* were significant (*p*-value = 3.18 × 10^−9^ and 1.22 × 10^−19^, respectively) in this GWAS dataset [40]. As shown in Figure 4a and Appendix A, after SMR and HEIDI analysis of the latest GWAS data, the *STX4* and *BST1* genes were still in the intersection range. These results not only provide trustworthy evidence for the newly discovered *STX4* and *BST1* genes but also demonstrate the reliability of the six PD risk genes found in this study.

As shown in Appendix A and Appendix A, the effect size of the six risk genes is summarized, indicating the effect and direction of the gene expression level on the risk of PD. When the effect value $b_{SMR}>0$, it means that the gene expression is positively associated with disease risk; $b_{SMR}<0$ means that the gene expression is negatively associated with disease risk; $b_{SMR}=0$ means that the gene expression is irrelevant. For example, *BST1* is a risk gene located on Chr4 (FDR P_SMR_ = 0.0214, P_HEIDI_ = 0.738), and its corresponding eQTL probe has a significant *p*-value of 1.01 × 10^−229^ (Figure 5a). In addition, according to the effect value of eQTL ($b_{zx}$ = 0.58) and the effect value of the SNP in the GWAS of the phenotype ($b_{zy}$ = −0.14), SMR estimates that $b_{xy}$ = −0.24 can represent no confounding effect of gene expression on PD (Figure 5b). A negative estimate of the effect of gene expression on PD (b_SMR_ = −0.24) indicates the inhibitory effect of *BST1* expression.

### 3.3. Elucidating the Pathogenesis of Two Risk Genes

We combined GWAS, eQTL and mQTL data to conduct a pairwise analysis to explore whether there are consistent signals at multi-omics levels (Figure 3). In the first step, significant eQTL data were used as an instrumental variable combined with GWAS data to identify a total of six risk genes associated with PD. The second step was to calculate the causality between DNAm and its neighboring genes (within 2 Mb of each DNAm probe). The third step was to perform pleiotropic association analysis between the DNAm site and the meta-analysis GWAS data to prioritize the DNAm loci.

Using aggregated mQTL data compiled by Wu et al. [10], we screened for significant SNPs (*p*-value < 5 × 10^−5^) within 2 Mb of each probe as instrumental variables with at least one cis-mQTL at *p*-value < 5 × 10^−8^. As shown in Table 2 and Figure 3, after mQTL-eQTL analysis, a total of 51,940 DNAm probes were identified as being related to eQTL probes. In the DNAm-PD analysis, 24 PD-related DNAm probes were detected, of which 13 DNAm probes were not rejected by HEIDI (Table 2 and Figure 3).

To explore whether the DNAm locus is in the regulatory region of genes, we used 14 major functional annotation categories from the REMC to functionally locate DNAm probes [41] (Appendix A). Apparent functional annotation was performed on all 73,973 DNAm probes, and the enrichment of expression-related DNAm probes was analyzed using the chi-square test, including promoter (fold-change = 1.27, *p*-value = 1.01 × 10^−54^), strong transcription (Tx, fold-change = 1.32, *p*-value = 2.44 × 10^−27^), strong transcription and Enhancer (TxEn, fold-change = 1.31, *p*-value = 3.88 × 10^−28^) and other functions that promote expression. In addition, there was significant non-enrichment of these 51,940 DNAm probes in the transcriptional repression region, such as repressed polycomb (ReprPC, fold-change = 0.79, *p*-value = 1.32 × 10^−27^) and quites (fold-change = 0.70, *p*-value = 1.03 × 10^−98^) (Appendix A).

As shown in Figure 3, by combining the results of SMR analysis in paired omics, five genes (corresponding to eight DNAm probes and seven eQTL probes) with pleiotropic causality were found to produce strong signals of multiomics, including *C17ORF69 (CRHR1)*, *KIAA1267* (*KANSL1*), *LRRC37A4* (*LRRC37A*), *MGC57346* (*CRHR1*), *NSF* and *STX4*. Figure 6 shows the *p*-values of these genes in three omics analyses, and we can observe that the *C17ORF69* (*CRHR1*), *KIAA1267* (*KANSL1*), *LRRC37A4* (*LRRC37A*), *MGC57346* (*CRHR1*) and *NSF* genes on Chr17 are important. Appendix A shows a summary of the cross-omics correlation of *STX4* on Chr16.

We found that the two risk genes (*CRHR1* and *LRRC37A4*) in the analysis of mQTL-GWAS and mQTL-eQTL were significant and consistent in SNP-associated signals. Its corresponding DNAm locus belongs to the promoter functional region, which suggests a possible biological pathway for regulating disease risk (Table 2): (i) As shown in Figure 7a, the rs17426174 mutation changes the level of DNAm, and when the DNAm level (cg17117718) of the *CRHR1* promoter is low, the repressor binds to the promoter, thereby inhibiting the transcription of *CRHR1* (DNAm–gene effect value b_SMR_ = 0.33) and increasing the risk of PD (gene–PD effect value b_SMR_ = −0.51); (ii) as shown in Figure 7b, genetic mutations (rs11012 or rs17426174, respectively) regulate methylation probes (cg08113562 or cg17117718, respectively) to change the degree of DNAm. When the DNAm level of the *LRRC37A4* promoter is low, the transcription factor and promoter are combined, thereby promoting the expression of *LRRC37A4* (DNAm–gene effect value b_SMR_ = −0.1 and −0.09, respectively) and increasing the risk of PD (gene–PD effect value b_SMR_ = 1.58).

## 4. Discussion

This study explored the risk genes and pathogenesis of PD by integrating a meta-analysis of GWAS, eQTL and mQTL data. Four known genes, *CRHR1*, *KANSL1*, *NSF* and *LRRC37A*, and two new risk genes, *STX4* and *BST1*, were identified. Combined with the three omics datasets and the functional annotation of the DNAm locus, the underlying mechanism of two risk genes affecting PD risk was preliminarily clarified. Overall, this research identified the risk genes and potential pathogenesis that affect PD, which has significance for the diagnosis and treatment of PD.

The GWAS result was analyzed after the meta-analysis was used, which is a crucial step for subsequent SMR analysis. As shown in Figure 1a, we found that more than 1600 mutation sites in approximately 6 million genetic mutation sites were significantly related to PD, which improved the statistical power of this study. As shown in the quantile–quantile plot (Figure 1b), when the *p*-value < 1 × 10 ^−3^, the GWAS result of the meta-analysis is clearly deviated from the uniform distribution, which indicates that the genotype may be affected by natural selection [42]. GO enrichment revealed that four out of six significantly enriched biologically functional regions have been reported to be associated with PD (Figure 2).

Relevant cellular or animal experiments have provided reliable evidence for the identification of risk genes. Of the eight genes analyzed by SMR, *SNCA* [43,44], *CRHR1* [27,28], *KANSL1* [27], *LRRC37A* [29], *BST1* [30,31,32,33] and *DGKQ* [32] have been shown in previous studies to be related to PD at the genome-wide level (Table 1). Related reports have indicated that *CRHR1* has a neuroprotective effect [36]; a recent genetic analysis found that *BST1* is a susceptibility gene for PD [33], and previous experiments in *BST1* knockout mice suggested that the gene might be beneficial in promoting treatment for PD [34].

Identifying new drug targets through risk genes is a highly effective drug discovery strategy, which is conducive to drug repositioning research for PD [45]. To assess the potential value of risk genes in drug discovery, this paper obtained all drug target genes from the major drug database DrugBank, which includes drugs approved in clinical trials or experimental drugs [46]. *CRHR1* was found to overlap with the DrugBank drug target, which corresponds to hydrocortisone for Cushing’s disease [35]. The *CRHR1* receptor can be combined with CRF, the main point regulator of the hypothalamic–pituitary–adrenal axis, to regulate a large amount of hormone cortisol released downstream, indicating that *CRHR1* has a neuroprotective effect [36,37]. There is also evidence from animal experiments that hydrocortisone can stimulate the expression of Parkin through the CREB pathway, and the induced expression of Parkin is the cause of its neuroprotective effect [38]. Therefore, hydrocortisone, as a *CRHR1* repositioning drug, may become a potential drug in the treatment of PD.

There are also some limitations to this study. First, this study did not perform tissue-specific identification. The expression data we used were derived from blood [18]; it will be better to analyze the expression data from brain tissue. However, some studies have shown that genetic influences on eQTL or mQTL data are highly correlated between independent brain and blood samples [47,48]. Zhu et al. found that expression data from brain tissue or blood did not significantly affect the gene recognition of schizophrenia [10]. Second, the HEIDI test is too conservative [9]. Therefore, we adopted a multiple test method to reduce the threshold of the HEIDI test to reduce the effect of possible false negative genes (Table 1).

## 5. Conclusions

In summary, the results suggest that *CRHR1*, *KANSL1*, *NSF*, *LRRC37A* and two new genes *STX4* and *BST1* are related to the risk of PD. In particular, the discovery of the *CRHR1* target indicates that hydrocortisone may become a potential drug for the treatment of PD. In addition, the expression mechanism of *CRHR1* and *LRRC37A* also illustrates the important role of DNAm in gene expression and disease occurrence. This study provides a process for analyzing disease genes and pathogenesis, which is of great significance in clinical treatment.

## Figures and Tables

**Figure 1 genes-11-01100-f001:**
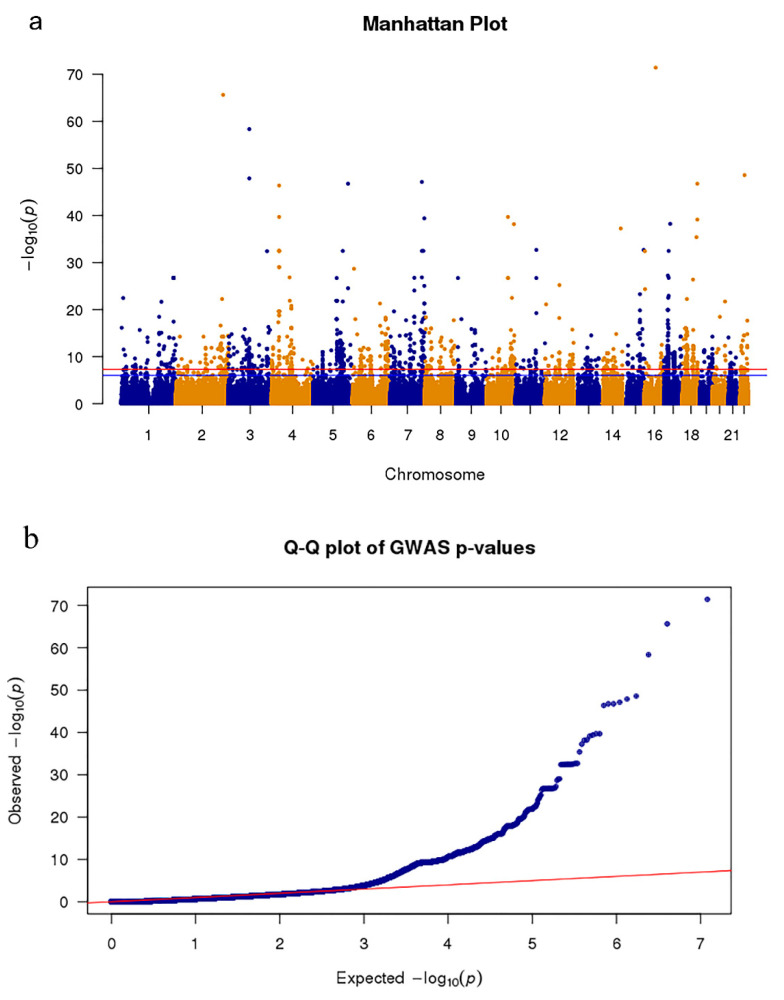
Meta-analysis Genome-wide association studies (GWAS) results show single nucleotide polymorphisms (SNPs) associated with Parkinson’s disease (PD). (**a**) The Manhattan plot presents the *p*-value distribution of SNPs. The red line represents the significance level (*p*-value = 7.83 × 10^−9^) across the genome. (**b**) Quantile–quantile plot tests of the effectiveness of SNPs. The *x*-axis and *y*-axis represent the −log10 (*p*-values) of the expected SNPs and true SNPs, respectively.

**Figure 2 genes-11-01100-f002:**
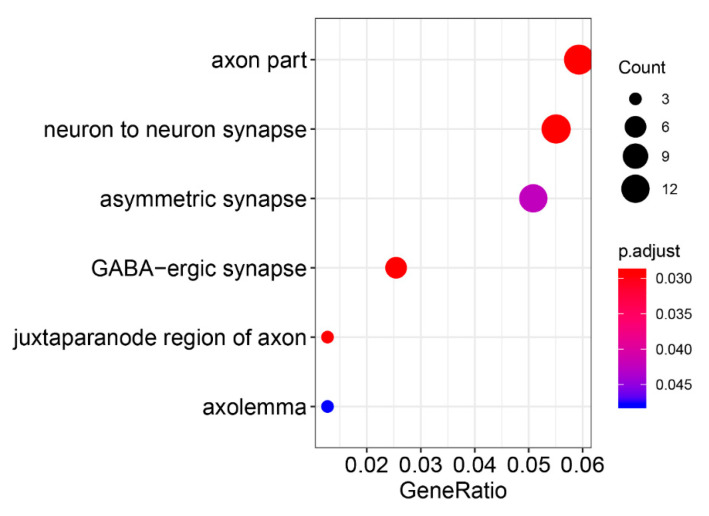
Gene Ontology enrichment of meta-analysis GWAS results.

**Figure 3 genes-11-01100-f003:**
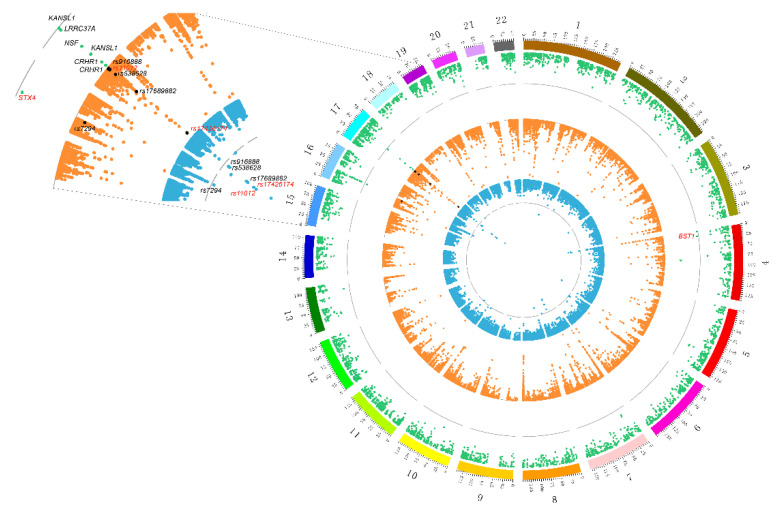
Prioritized risk genes and functional regions of PD with consistent association signals at the multiomics layers. The outermost green represents the results of expression quantitative trait locus (eQTL)-GWAS. Dots that exceed the gray line indicate that they have passed the SMR test. Marked dots represent six pleiotropic genes that passed the HEIDI test (black indicates four known genes; red indicates two new genes). The orange circle in the middle indicates the eQTL-methylation quantitative trait locus (mQTL) results. The innermost blue circle indicates the mutation site of mQTL-GWAS. In these two circles, the mutation sites that passed the SMR and HEIDI were marked, and the red-marked sites were potential mechanism regulatory sites.

**Figure 4 genes-11-01100-f004:**
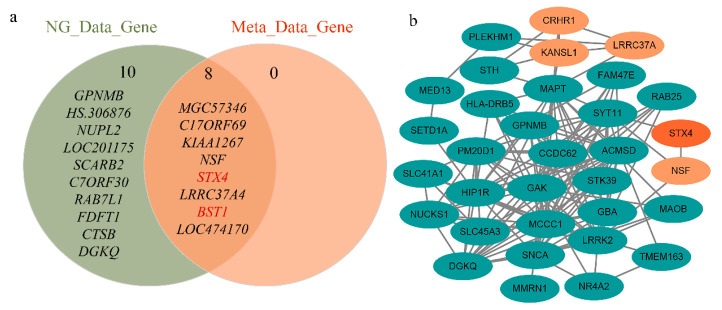
Independent dataset validation of six identified risk genes. (**a**) Significant genetic intersection of independent GWAS data and meta-analysis GWAS data tested. The most significant genes of the latest GWAS data and the significant genes after SMR and HEIDI include *STX4* and *BST1*, respectively. (**b**) Protein–protein interaction network of significant genes in the PD-Gene identified by SMR and HEIDI. *STX4* interacts with known PD risk genes.

**Figure 5 genes-11-01100-f005:**
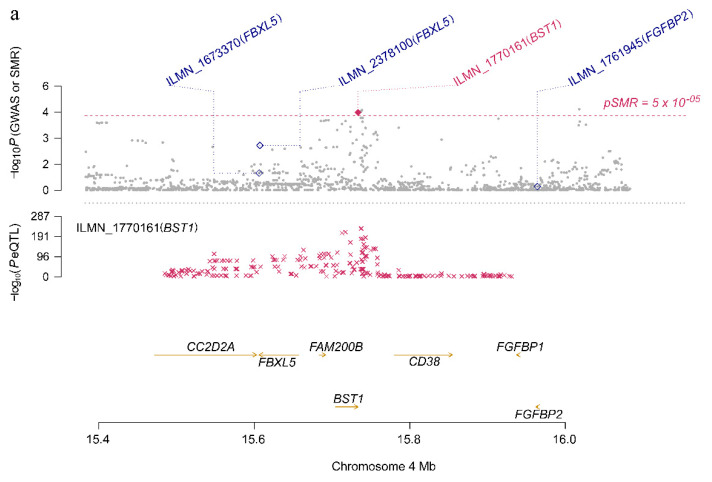
*BST1* locus for PD. (**a**) The abscissa represents the coordinate axis on the chromosome interval. In the top graph, the brown dots indicate the *p*-values of each SNP in the GWAS meta-analysis, and the diamonds indicate the *p*-values of the probes in the SMR test. The figure below refers to the *p*-value of the cis-eQTL-labeled *BST1* probe in eQTL data. *BST1* loci that passed the SMR and HEIDI tests are highlighted with red dots. (**b**) The effect sizes of SNPs from GWAS and eQTL effects of SNPs. The orange dotted line indicates the most significant cis-eQTL effect value $b_{xy}$, and $b_{xy}$ < 0 indicates that gene expression is negatively correlated with PD risk.

**Figure 6 genes-11-01100-f006:**
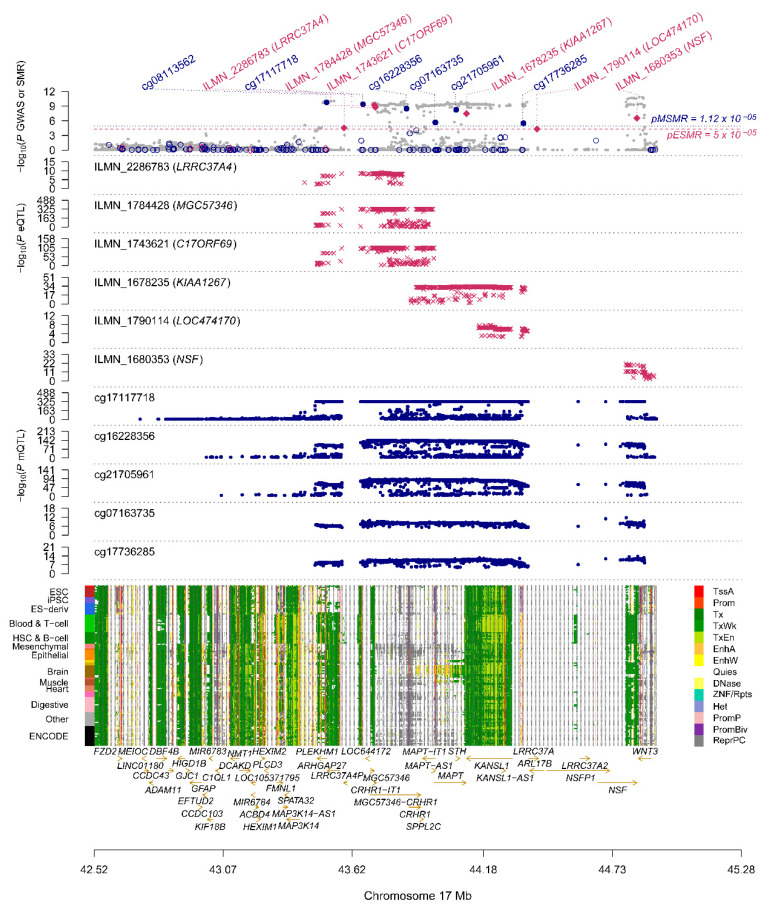
Summary of SMR correlation analysis across GWAS, eQTL and mQTL on Chr17. The figure above shows the −log10 (*p*-values) of the SNPs from meta-analytic GWAS data. The middle two graphs are −log10 (*p*-values) of SNPs in eQTL and mQTL, respectively. The lower diagram shows chromatin status annotation information.

**Figure 7 genes-11-01100-f007:**
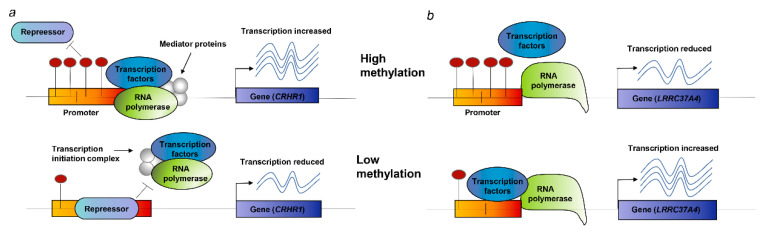
Potential regulatory mechanisms. (**a**) When the DNAm site in the promoter is high, the binding of the inhibitor and the promoter is prevented; the transcription factor normally binds to the promoter, and the expression of *CRHR1* increases; when the DNAm site is low, the inhibitor binds to the promoter, and *CRHR1* transcription is suppressed. (**b**) When DNAm in the promoter is high, the binding of transcription factors is disrupted, thereby suppressing the expression of *LRRC37A4*. When DNAm is low, transcription factors usually bind to the promoter, and the expression level of *LRRC37A4* increases.

**Table 1 genes-11-01100-t001:** Summary of risk genes identified by the summary data-based Mendelian randomization (SMR) and heterogeneity in dependent instruments (HEIDI) methods for PD.

ProbeID	Chr	Gene	topSNP	A1	P_GWAS_	FDR P_SMR_	P_HEIDI_
ILMN_1784428	17	*MGC57346* *(CRHR1)*	rs4074462	T	4.2 × 10^−10^	3.91 × 10^−6^	0.605
ILMN_1743621	17	*C17ORF69* *(CRHR1)*	rs17426195	A	3.53 × 10^−10^	4.91 × 10^−6^	0.507
ILMN_1678235	17	*KIAA1267* *(KANSL1)*	rs4630591	T	5.59 × 10^−10^	6.43 × 10^−5^	0.989
ILMN_1680353	17	*NSF*	rs183211	A	1.24 × 10^−9^	3 × 10^−4^	0.036
ILMN_1680313	16	*STX4*	rs8056842	C	1.39 × 10^−5^	2.14 × 10^−4^	0.732
ILMN_2286783	17	*LRRC37A4* *(LRRC37A)*	rs1635298	T	1.56 × 10^−8^	2.14 × 10^−4^	0.225
ILMN_1770161	4	*BST1*	rs4698412	G	2.83 × 10^−5^	2.14 × 10^−4^	0.738
ILMN_1790114	17	*LOC474170* *(KANSL1)*	rs9915547	C	2.85 × 10^−9^	2.48 × 10^−4^	0.551

**Table 2 genes-11-01100-t002:** Gene loci that pass SMR and HEIDI at the level of DNAm, transcript and trait.

Methylation (M)—Trait (T)				Methylation—Expression (E)			Expression—Trait	
Mprobel_Region	Mprobe	M_SNP	MT_b_SMR_	Eprobe	E_SNP	ME_b_SMR_	Gene	ET_b_SMR_
Prom	cg17117718	rs17426174	−0.17	ILMN_1743621	rs17426174	0.33	*C17ORF69* *(CRHR1)*	−0.51
ReprPC	cg16228356	rs17689882	0.27		rs17689882	−0.54		
TxWk	cg21705961	rs17689882	0.34		rs17689882	−0.67		
TssA	cg07163735	rs538628	−0.97		rs17689882	2.25		
Quies	cg17736285	rs916888	0.88		rs1876831	−1.88		
Prom	cg08113562	rs11012	−0.22	ILMN_1678235	rs16940665	−0.26	*KIAA1267* *(KANSL1)*	0.88
Prom	cg17117718	rs17426174	−0.17		rs16940665	−0.18		
ReprPC	cg16228356	rs17689882	0.27		rs17689882	0.31		
TxWk	cg21705961	rs17689882	0.34		rs17689882	0.38		
Quies	cg17736285	rs916888	0.88		rs17575507	1.07		
Prom	cg08113562	rs11012	−0.22	ILMN_2286783	rs11012	−0.10	*LRRC37A4* *(LRRC37A)*	
Prom	cg17117718	rs17426174	−0.17		rs17426174	−0.09		1.58
ReprPC	cg16228356	rs17689882	0.27		rs17426174	0.15		
TxWk	cg21705961	rs17689882	0.34		rs17426174	0.19		
TssA	cg07163735	rs538628	−0.97		rs17426174	−0.63		
Quies	cg17736285	rs916888	0.88		rs17426174	0.54		
Prom	cg17117718	rs17426174	−0.17	ILMN_1784428	rs17426174	0.57	*MGC57346* *(CRHR1)*	−0.3
ReprPC	cg16228356	rs17689882	0.27		rs17689882	−0.93		
TxWk	cg21705961	rs17689882	0.34		rs17689882	−1.16		
TssA	cg07163735	rs538628	−0.97		rs17689882	3.91		
Quies	cg17736285	rs916888	0.88		rs1876831	−3.27		
Prom	cg08113562	rs11012	−0.22	ILMN_1680353	rs199448	0.17	*NSF*	−1.09
Prom	cg17117718	rs17426174	−0.17		rs199448	0.12		
ReprPC	cg16228356	rs17689882	0.27		rs199448	−0.2		
TxWk	cg21705961	rs17689882	0.34		rs538628	−0.25		
TssA	cg07163735	rs538628	−0.97		rs538628	0.62		
Quies	cg17736285	rs916888	0.88		rs916888	−0.62		
Tx	cg01067137	rs7294	0.29	ILMN_1680313	rs7294	−0.57	*STX4*	−0.45

Mprobel_region = physical positions of methylation probes; Mprobe = ID of methylation probes; M_SNP = top significant SNP of methylation probes; MT_b_SMR_ = SMR-estimated effect of DNA methylation on phenotype; ME_b_SMR_ = SMR-estimated effect of DNA methylation on gene expression; ET_b_SMR_ = SMR-estimated effect of gene expression on PD.

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
