# Peer review of "Identifying Risk Genes and Interpreting Pathogenesis for Parkinson’s Disease by a Multiomics Analysis"

_genes, 2020, doi:10.3390/genes11091100_

Round 1
Reviewer 1 Report
In this study Wen-Wen Cheng and collaborators perform a GWAS meta-analysis combined with expression data to identify risk genes associated with Parkinson’s disease (PD). Results obtained identified 6 genes (CRHR1, KANSL1, NSF, LRRC37, STX4 and BST1) that shown to be related with PD increasing their significance towards future diagnostic and novel therapies.
The manuscript presents a moderate novelty in the field, but it is well structured and the methodology, as well as the experimental design, is correct. However, there are some elements that can be revised to clarify specific aspects and improve the document:
Authors indicated in the abstract as well as in results section 3.2. and discussion that the analysis identifies two new risk genes related with PD. It is true that there is no previous evidence abut the relation between STX4 and PD. However, previous studies indicate that BST1 could be associate with an increase in the risk of PD (see as an example: Li et al., 2019, Neuropsychiatr Dis Treat. 2019 Apr 30;15:1089-1102. doi: 10.2147/NDT.S190935. eCollection 2019).
Bibliography associated to the genes founded in this study should be revised and update.
Quality of figures 3, 4, 5 & 6 must be improved to help visualize the results. Resolution of colors and words looks very low.
Reviewer 2 Report
Parkinson’s disease (PD) is the second most common degenerative neurological disorder after Alzheimer’s disease. It is estimated that PD affects 1 percent of the population over the age of 60. A small percentage of people with PD (4 percent of all cases) are diagnosed before the age of 50. Overall, as many as 1 million Americans are living with PD, and approximately 60,000 Americans are diagnosed with PD each year. In the manuscript, Cheng and colleagues explored the risk genes and pathogenesis of PD by integrating a meta-analysis of GWAS, eQTL and mQTL data. Interestingly, the Authors identified four known genes, CRHR1, KANSL1, NSF and LRRC37A, and two new risk genes, STX4 and BST1.
Overall, the manuscript is really interesting, complete and have a critical point of view; the identification of the risk genes is important for PD future therapeutic approaches.
However, there are minor flaws:
- The Authors stated: “Therefore, hydrocortisone, as a CRHR1 repositioning drug, may become a potential drug in the treatment of PD.” (LINE 310) indicating hydrocortisone as a potential drug in the PD treatment, however, it is well-know and there is an already published study showing the effectiveness of hydrocortisone in a vivo PD model (https://doi.org/10.1038/s41598-017-00614-w). Thus, the Authors should at least cite in the manuscript this work.
- The introduction seems too concise, please increase the introduction part
- Bibliographic references are quite old, they should be updated with some more recent ones
